# Study on Radio Frequency-Treated Agricultural Byproducts as Media for *Hericium erinaceus* Solid-State Fermentation for Whitening Effects

Zih-Yang Lin, Chia-Ling Yen and Su-Der Chen *

Department of Food Science, National Ilan University, Number 1, Section 1, Shen-Lung Road, Yilan County, Yilan City 260007, Taiwan; jjj71996@gmail.com (Z.-Y.L.); jonahyen99@gmail.com (C.-L.Y.)
* Correspondence: sdchen@niu.edu.tw; Tel.: +886-9317761; Fax: +886-39351892

**Abstract:** Hot air-assisted radio frequency (HARF) is considered a rapid heating process. In order to improve the circular economy of agricultural byproducts, this study used different proportions of HARF stabilized rice bran (R) from milling rice, HARF dried ginseng residue (G) from ultrasonic extraction, and peanut residue (P) from HARF roasting and oil extraction as the *Hericium erinaceus* solid-state fermented media. Then, the whitening effects of water extracts from media and fermented products were analyzed. First, the surface temperature of 1 kg rice bran exceeded 90 °C after 3 min of 5 kW HARF heating, effectively deactivating lipase. The combinations of 1 kg of rice bran with 0.5, 1, 1.5, and 2 kg of ginseng residue (85% moisture content) were dried using 5 kW HARF. Each of the drying rates was about 27 g/min, and the drying periods were 14, 30, 46, and 62 min, respectively, which were used to reduce the moisture content below 10%. Compared to traditional air drying for ginseng residue, HARF drying may save up to 96% of time and 91% of energy consumption. Then, the ratio of dried R, G, and P was 4:1:1, mixed with 45% moisture as solid-state media for *Hericium erinaceus* and 5 weeks of cultivation at 25 °C. In comparison to the control group, the water extracts at 100 μg/mL from media R4G1, R4G1P1, and fermented HER4G1P1 products exhibited tyrosinase inhibition of 29.7%, 52.4%, and 50.7%, respectively. These extracts also reduced the relative melanin area of 78 hpf zebrafish embryos by 21.57%, 40.20%, and 58.03%, respectively. Therefore, HARF can quickly dry agricultural byproducts as media for *Hericium erinaceus* solid-state fermentation while also providing a significant whitening effect for cosmetic applications.

**Keywords:** radio frequency (RF); drying; agricultural byproducts; fermentation; whitening

## 1. Introduction

*Panax ginseng* C.A. Meyer, commonly known as ginseng, is a popularly used traditional medicinal herb in Asian countries. Several bio-functional effects of *P. ginseng* have been described, including immunomodulatory activity, cardiovascular system protection, nutritional fortification, improvements of liver function, anti-diabetes, anti-cancer, anti-apoptotic, anti-inflammatory, and antioxidant activities [1–3]. In recent years, ginseng has been demonstrated to effectively treat skin hyperpigmentation disorders by suppressing melanogenesis and melanosome transport [4]. Additionally, ginseng has shown promise in altering signaling factors in skin tissue, resulting in anti-UV, anti-skin cancer, and anti-hair loss therapeutic benefits [1]. Therefore, ginseng can be a valuable ingredient in skincare products.

Several popular ginseng beverages are available on the market, and these products are typically manufactured using ginseng extraction or boiling methods. If ginseng residues can be efficiently dried via HARF drying, it will address the waste disposal issue, and these dried ginseng residues could potentially be used as solid-state fermented media or ingredients in skin-whitening patches or cosmetics.

Rice bran, a by-product from rice milling, is primarily used as animal feed, and a small percentage is processed into rice bran oil or other products. However, rice bran contains lipase and lipoxygenase, which have a risk of lipid hydrolysis and rancidity. Methods for stabilization of rice bran to deactivate enzymes include chemical stabilization, cold storage, hydrothermal treatment, and steam treatment [5]. A 5 kW RF heating process was employed to stabilize rice bran. It only took 2 min to heat the 1 kg of rice bran from 20 °C to 100 °C, and the temperature distribution was uniform. The RF-treated rice bran could be stored for up to 8 weeks at 4, 25, and 37 °C without experiencing rancidity [6].

Water extracts from rice bran were shown to be more effective at inhibiting tyrosinase activity than ethanol extracts. At a concentration of 50 µg/mL, the water extracts inhibited tyrosinase by 30%, and at 200 µg/mL, it reached 27.8% inhibition. Furthermore, ethanol extracts at doses of 50, 100, 150, and 200 µg/mL inhibited tyrosinase activity by 7.2%, 13.82%, 16.19%, and 19.91%, respectively [7]. Jun et al. [8] previously demonstrated that $\gamma$-oryzanol inhibits tyrosinase activity in a concentration-dependent manner, and the melanin content in B16F1 cells was significantly and dose-dependently reduced ($-13\%$ and $-28\%$ at 3 and 30 µM $\gamma$-oryzanol, respectively). Therefore, the water extracts from rice bran exhibit significant tyrosinase inhibitory properties, making them promising candidates for skin-whitening treatments.

The germination of peanut kernels increased the resveratrol, GABA, flavonoid and polyphenol contents, and the ability to scavenge DPPH [9]. Resveratrol could inhibit tyrosinase activity and decrease the melanin content in 78 hours post-fertilization (hpf) in zebrafish embryos [10]. Peanut shell extract, using ultrasound-assisted extraction, had high antioxidant activity and excellent skin-whitening and anti-wrinkle effects, giving peanut shell great potential as a natural cosmetic and food ingredient [11]. Peanut is one of the major oilseeds with oil content in the range of 47~50%, and peanut oil obtained from the roasted kernel (170 °C for 20~30 min or 180~190 °C for 25~30 min) has inactivated enzyme and protein coagulation in the oil and facilitates the release of oil constituents during the extraction process [12]. The peanut residue after oil expression is greater than 50%, which can be utilized as a solid-state fermented medium rather than feed.

*Hericium erinaceus*, commonly known as Lion's Mane mushroom, is a fungus with both culinary and medical uses. Its fruiting bodies and mycelia exhibit various bioactive components, such as polysaccharides, triterpenoids, erinacines, flavonoids, and polyphenols [13]. It has several bio-functional activities, such as immunomodulation, antitumor, antioxidant, neuroprotection, and enhancing memory. The 200 ppm of ethanol and water extracts from *Hericium erinaceus* solid-state fermented product had significant protective and repairing effects on 1% ethanol damage to neural cells in zebrafish embryos [14].

Dielectric heating methods are utilized in both RF (radio frequency, RF) and microwave heating. As RF or microwave energy passes through a food substance, water molecules encounter dipolar friction, and ions rapidly migrate. This results in significant vibration to generate heat, leading to rapid and uniform internal heating of the food. RF heating has been found applications in various agricultural processes such as pest control, pasteurization, thawing, enzyme inhibition, and drying [15]. HARF drying not only expedites the drying process but also ensures a more even temperature distribution within the samples. RF has a much greater penetration depth compared to microwaves, making it particularly suitable for large-scale food industry production [16].

When drying 1 kg, 1.5 kg, and 2 kg of soybean residue using 45 °C hot air-assisted 5 kW RF, it took 14, 22, and 30 min to reduce the moisture content from 80% to 18%. The drying rates were 54.7, 52.3, and 50.7 g of water/min, respectively, demonstrating a constant drying period. The energy usages were 1.47 kWh, 2.31 kWh, and 3.15 kWh, respectively. In contrast, drying with only 45 °C hot air took 330 min, leading to a high energy consumption of 13.53 kWh [17].

However, HARF drying of high-moisture products often leads to excessive RF induction power, and it is a bottleneck in HARF operation. Huang et al. [18] employed a solution containing bitter melon extract with soybean fiber powder to relieve the issue of excessive

moisture content in food and accelerate RF drying, thus achieving time and energy savings. In this research, low-moisture rice bran was employed as a drying aid in high-moisture ginseng residue to optimize the RF drying process.

Skin pigmentation is caused by an excessive synthesis and accumulation of melanin. Tyrosinase initiates the melanogenesis process by oxidizing L-tyrosine and L-DOPA to dopaquinone. Melanin production can be effectively suppressed by inhibiting these enzymes and proteins involved in melanin synthesis or by using reducing agents to convert dopaquinone back to DOPA [19]. With the increasing awareness of animal protection and welfare, scholars have proposed the 3R principles: Replacement, Reduction, and Refinement. Several countries have also enacted legislation to prohibit the use of animals for cosmetics testing.

Zebrafish (*Danio rerio*) have become a popular study model due to their advantages, such as transparent embryos and the ease with which developing organs can be observed. All vertebrate characteristics of animals can be observed within approximately 2~3 days [20]. Strähle et al. [21] discovered that zebrafish become independent feeders at around 120 hpf. This timeframe aligns with the European Union Directive 2010/63/EU, which excludes laboratory animals in the early stages of the definition. Therefore, zebrafish embryos within the first five days of fertilization are not protected by animal welfare standards around the world. In recent years, zebrafish animal models have gained popularity for investigating the efficacy and safety of cosmetic chemicals.

The 30 mg/mL 80% ethanol extracts from *Saccharomyces cerevisiae* fermented rice could reduce the melanin content to 71% through tyrosinase-mediated MITF downregulation on B16 cells and in vivo zebrafish embryos (48 hpf) [22]. Ginsenosides in methanol extract from *Panax ginseng* [3,23,24] or American ginseng could reduce the melanin content and tyrosinase activity in zebrafish embryos [25]. Zebrafish embryos were immersed in black ginseng extract for 72 h, and a decrease in melanin depth on the zebrafish skin surface was found as the treatment concentration rose [2].

Therefore, the objectives of this study were to (1) enhance the usability of agricultural byproducts such as rice bran and ginseng residue via HARF treatment; (2) create a solid-state medium for *Hericium erinaceus* from a mixture of RF-dried ginseng residue and rice bran, along with de-oiled peanut residue with 45% moisture; and (3) analyze their inhibitory effect on tyrosinase activity. Additionally, the whitening effect of the extracts was evaluated using zebrafish embryos.

## 2. Materials and Methods

### 2.1. Materials

The ginseng residue with 85% moisture content after ultrasonic extraction was provided by ACEXTRACT Biotechnology Co., Ltd. (Yilan, Taiwan). Indica rice bran of Taichung No. 10 was obtained from Minfeng Organic Farm (Yilan, Taiwan), with initial moisture content of around 14%. The peanuts used in this experiment were purchased from Tainan No. 12, Zhuangwei Township, Yilan County. *Hericium erinaceus* (BCRC 36470) was purchased from the Bioresource Collection and Research Center (Hsinchu, Taiwan). Magnesium sulfate (MgSO$_4$·7H$_2$O), dipotassium hydrogen phosphate (K$_2$HPO$_4$), and glucose were purchased from Wako Pure Chemical Industries, Ltd. (Osaka, Japan). Yeast extract and potato dextrose agar medium (PDA) were purchased from Difco Co. (Sparks, MD, USA). Kojic acid, L-tyrosine, arbutin, and tyrosinase were purchased from Sigma Chemical Company (St. Louis, MO, USA). Sodium dihydrogen phosphate (NaH$_2$PO$_4$) and sodium dihydrogen phosphate (Na$_2$HPO$_4$) were purchased from WAKO Pure Chemical Industries, Ltd. (Osaka, Japan). Wild-type zebrafish (AS-AB) was purchased from the Taiwan zebrafish center branch (TZCNHRI, Miaoli, Taiwan) at National Health Research Institutes. The ELISA 96-well microplate (90015-2NB) was purchased from Alpha Plus Scientific Corp. (Taoyuan, Taiwan). Tissue PE LB™ was purchased from G-Biosciences (Maryland Heights, MO, USA). A 0.22 μm syringe filter was purchased from Advangene Consumables (Lake Bluff, IL, USA).

### 2.2. Equipment

Equipment used included hot air-assisted radio frequency (HARF) equipment (as shown in Figure 1) (5 kW, 40.68 MHz, 220V, Yh-Da Biotech Co., Ltd., Yilan, Taiwan), infrared thermometer (Testo 104-IR, Hot Instruments Co., Ltd., New Taipei, Taiwan), electric oven (Channel DCM-45, Sci-Mistry Co., Ltd., Yilan, Taiwan), precision balance (HDW-15L, HengSin Measurement Technology Co., Led., Yilan, Taiwan), and high-speed grinder (RT-04, Sci-Mistry Co., Ltd., Yilan, Taiwan). The microwave extraction device in our laboratory was self-assembled, including an adjustable microwave power microwave (Yen Hua Biotech Co., Ltd., Tainan City, Taiwan), consisting of a microwave oven with a flat top and a flask condenser with a volume of 500 mL. Then, a 4 °C cold water cycle machine (Firstek, B8402H, One Liter Technology Co., Ltd., New Taipei City, Taiwan) was added to maintain the condenser at a low temperature. We also used a high-speed batch top centrifuge (Z300, Hermle AG, Gosheim, Germany), vortex mixer (Vortex Mixer VM-2000, Shin Kwang Machinery Industry Co., Ltd., New Taipei, Taiwan), vacuum concentrator (Eyela rotary evaporator N-1000, Tokyo Rikakikai Co., Ltd., Tokyo, Japan), ultrasonic cleaner (DC-600H, Macro Fortunate Co., Ltd., New Taipei, Taiwan), microplate spectrophotometer (ChroMate 4300, Tseng Hsiang Life Science Ltd., Taipei, Taiwan), pH meter (6177M, Jenco Electronics, Ltd., Taipei, Taiwan). Stereo microscope (SMZ745T, Sage Vision Co., Ltd., New Taipei, Taiwan), 2 L spawning box (Taikong Corporation, New Taipei, Taiwan), 25 °C incubator (LM-600R, Yihder Co, Ltd., New Taipei, Taiwan), cold press oil press (VGM-220, Songyu Electric Co., Ltd., Miaoli, Taiwan), high-temperature steam vertical autoclave (Tommy SS-325, Tokyo, Japan), and vertical laminar flow clean bench (LAB-3A, Sage Vision, Co. Ltd. New Taipei, Taiwan).

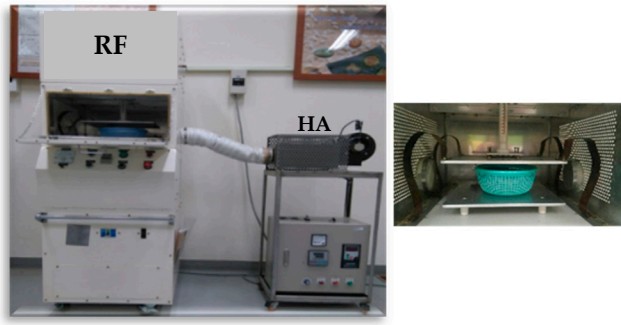

**Figure 1.** Hot air-assisted radio frequency (HARF) equipment (5 kW, 40.68 MHz).

### 2.3. Experimental Design

The experimental design diagram (Figure 2) was separated into two parts. First, the drying rate and energy consumption of HARF drying ginseng residue and rice bran combinations were assessed. The HARF-treated rice bran, ginseng residue mixture, and peanut residue after oil extraction were mixed in a 4:1:1 ratio (45% moisture content) to make a solid-state fermented medium. The water extracts from 5-week *Hericium erinaceus* solid-state fermented products were analyzed for the whitening effect.

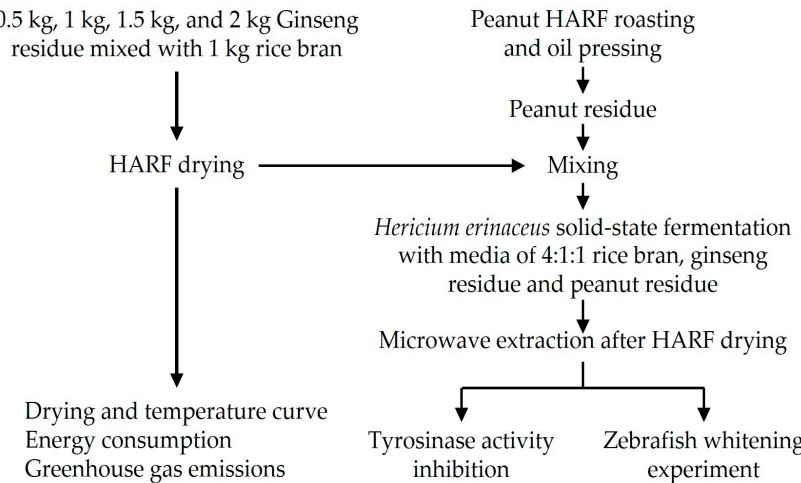

**Figure 2.** The experimental design diagram.

### 2.4. Rice Bran via RF Heating for Lipase Inactivation

The rice bran initially comprised approximately 14%. Subsequently, 1 kg of rice brain was placed in a PP box (length × width × height = 37.6 × 26.5 × 8.6 cm in dimensions). Utilizing 100 °C hot air-assisted 5 kW RF with electrode gaps of 10 cm, the rice bran was heated for 3 min until reaching 90 °C, effectively deactivating lipase, and the final moisture content of rice bran was reduced to 4%.

### 2.5. Preparation of Ginseng Residue and Rice Bran Mixture

The 1 kg of rice bran was uniformly combined with different quantities of ginseng residue (e.g., 0.5 kg, 1 kg, 1.5 kg, or 2 kg) before being placed into a PP container (38.5 × 31 × 10 cm in dimensions) for subsequent use.

### 2.6. HARF Drying Ginseng Residue and Rice Bran Mixture

Various combinations of the mixtures, consisting of 1 kg of rice bran mixed with 0.5 kg, 1 kg, 1.5 kg, and 2 kg of ginseng residue, underwent drying via the HARF (100 °C hot air with 5 kW RF) with electrode gaps ranging from 12 to 17 cm to read current (A) and calculate the RF power using the following equation:

$$\text{RF power output (kW)} = A \times \left(\frac{5}{1.6}\right) \tag{1}$$

Then, the loading mixtures were dried by HARF with a suitable gap to produce a final moisture content below 10%. During the drying process, the weight and temperature of the samples were measured, and these data were utilized to construct heating and drying curves. These curves facilitated the determination of heating rate and drying kinetics.

### 2.7. Determination of Energy Consumption for Ginseng Residue and Rice Bran Mixture

We utilized a clamp meter to measure the radio frequency equipment (three-phase electrical power) and cold-air drying equipment (single-phase electric power), recording the current in ampere (A) [6]. The energy consumption was calculated using the following equation:

$$\text{RF drying energy consumption (kWh)} = 220 \times \sqrt{3} \times A/1000 \times \text{drying time (h)} \tag{2a}$$

$$45\ °C\ \text{Cold air drying energy consumption (kWh)} = 220 \times A/1000 \times \text{drying time (h)} \tag{2b}$$

*2.8. Calculation of Greenhouse Gas Emissions for Ginseng Residue and Rice Bran Mixture*

We multiplied the 2022 electricity emission coefficient, as provided by the Environmental Protection Administration. Executive Yuan, R.O.C. (Taiwan) [26], by the energy consumption to compute the greenhouse gas emissions [27]. The greenhouse gas emissions could be determined using the following equation:

$$\text{The green house gas emissions (kgCO}_2\text{e)} = \text{electricity emission coefficient (0.495kgCO}_2\text{e/kWh)} \times \text{energy consumption (kWh)} \tag{3}$$

*2.9. Preparation of Peanut Residue*

Referring to the method outlined by Hung and Chen [28] for roasting peanuts and cold pressing oil, 1 kg of peanuts was placed in a polypropylene basket (width: 23.3 cm, height: 8.3 cm in dimension). The peanuts were roasted for 4.5 min using a 5 kW, 40.68 MHz hot air-assisted radio frequency with an electrode gap of 12 cm, achieving surface temperatures of 100 °C and interior temperatures of 120 °C. Subsequently, the roasted peanuts were then wrapped in parchment paper and divided into two packages before being loaded into the oil press. Oil pressing was performed at 45 °C, employing weights of 100, 200, 300, and 500 kg in four stages, each lasting 10 min. The peanut residue was then collected as a solid-state fermented medium.

*2.10. Hericium erinaceus Solid-State Fermentation*

2.10.1. Pre-Activation of *Hericium erinaceus*

Following the method outlined by Sun et al. [14], *Hericium erinaceus* was cultured on PDA at a constant temperature of 25 °C. Once the mycelium was grown, three pieces of 1 cm$^2$ *Hericium erinaceus* were taken and placed in the pre-activation solution (composed of 5% glucose, 0.1% yeast extract, 0.05% dipotassium hydrogen phosphate, 0.05% magnesium sulfate, and 150 mL RO water), where they were cultivated at a constant temperature of 25 °C and a rotation speed of 150 rpm for 7 days.

2.10.2. Preparation of PP Package and Cultivation of *Hericium erinaceus*

Referring to the method outlined by Sun et al. [14], a 500 g PP package containing a mixture of rice bran, ginseng residue, and peanut residue at a ratio of 4:1:1, with a moisture content of 45%, was placed in an autoclave for sterilization at 121 °C for 1 h. Subsequently, 10 mL of preactivated solution of *Hericium erinaceus* was withdrawn and added into PP package, thoroughly mixed, and then transferred to a 25 °C thermostat incubator for 5-week cultivation period.

2.10.3. Temperature and Drying Curve of HARF Dried *Hericium erinaceus* Solid-State Fermented Product

Referring to the method described by Yen and Chen [29], 1 kg of *Hericium erinaceus* solid-state fermented product was placed in a PP basket (38.5 cm length, 31 cm width, and 10 cm height in dimensions). The product underwent HARF drying with a 13 cm electrode gap until the moisture content was reduced to below 10%. Throughout the drying process, the weight and temperature changes of the mixture were recorded every 2 min. Temperature readings were taken at three points, and the average temperature was calculated. The dry basis moisture content was computed based on the change in weight of the mixture. Subsequently, dry basis moisture content (g water/g dry material) and temperature were plotted against time.

$$\text{Dry basis moisture content, MC(d.b.)} = (W_t - W_o)/W_o \tag{4}$$

where $W_t$ is the weight of the sample at time t, and $W_o$ is the dry weight of the sample.

### 2.11. Microwave Extraction

Then, 2.5 g of material was mixed with 50 mL of water, maintaining a solid to liquid ratio of 1:20, and subjected to extraction using a 300 W microwave for 5 min [30]. Following extraction, the solution was centrifuged, and the supernatant was isolated for subsequent experimentation.

### 2.12. Tyrosinase Activity Inhibition

Tyrosinase was dissolved in 0.1 M of phosphate buffer (pH = 6.8) to achieve a final concentration of 250 units/mL. Additionally, 30 mg of L-tyrosine was dissolved in 100 mL of 0.1 M phosphate buffer (pH = 6.8) to reach a final concentration of 300 mg/mL. The reaction was started by adding the following components into an ELISA 96-microplate: (A) 120 μL of 0.1M PBS + 40 μL of tyrosinase (250 unit/mL), (B) 160 μL of PBS, (C) 80 μL of PBS + 40 μL of tyrosinase + 40 μL of sample, and (D) 120 μL of PBS + 40 μL of the sample. The mixture was then incubated at room temperature (25 °C) for 10 min, followed by the addition of 40 μL of 300 mg/mL of L-tyrosine. Subsequently, after a further 10 min incubation at room temperature, the production of L-dopachrome was detected at 475 nm [31]. The percentage of mushroom tyrosinase inhibition was calculated by the following equation:

$$\text{Tyrosinase inhibition } (\%) = ((A - B) - (C - D))/(A - B) \times 100\% \tag{5}$$

### 2.13. Zebrafish Whitening Experiment

2.13.1. Breeding and Fertilized Zebrafish Egg Collection

The breeding and collection of fertilized zebrafish eggs were conducted as follows: wild-type zebrafish were raised in a water fertility system, and male and female fish were placed in separate tanks, each tank accommodating a maximum of 12 zebrafish. The water temperature was maintained at $28 \pm 2$ °C, and the photoperiod was utilized to induce zebrafish spawning, consisting of 14 h of light and 10 h of darkness. Male and female fish were placed together in the same small spawning box the day before egg collection, just before the light was turned off. Fertilized eggs were then collected the following day after natural mating occurred [32].

2.13.2. Whitening Effect of Zebrafish Embryos

Each group consisted of 50 zebrafish embryos at 6 h post-fertilization (hpf). These embryos were submerged in a 100 μg/mL test sample solution and left to soak for 72 h (until 78 hpf). Following soaking, the embryos underwent three rinses with reverse osmosis (RO) water after soaking. Subsequently, photographs were taken to obtain image analysis, whereas the group treated with 100 μg/mL of kojic served as the positive control [10].

2.13.3. Image Analysis of Zebrafish Embryos

The analysis was conducted using Image J image analysis software (version 1.54f, 2023). Color thresholding was executed in the HSB color mode, selecting melanin patches on the zebrafish body with brightness values ranging from 0 to 150. Notably, melanin spots on the zebrafish eyes were omitted from this study. Employing this method facilitated the assessment of alteration in melanin area on the zebrafish body surface in the images.

### 2.14. Statistical Analysis

The experimental results were presented as mean $\pm$ standard deviation (SD). A one-way analysis of variance (ANOVA) was performed and subsequently subjected to Duncan's multiple range tests of treatment mean by using Statistical Analysis System (SAS 9.4, SAS Institute, Cary, NC, USA), and the significant level was set at 0.05.

## 3. Results and Discussion

### 3.1. HARF Stabilizing Rice Bran and Drying Ginseng Residue

Rice bran, containing lipase and lipoxygenase, increases the risk of lipid degradation and rancidity. Therefore, heat stabilization is essential for preventing rancidity in the oils [5]. In this study, a 5 kW HARF system was used to heat 1 kg of rice bran for 3 min. As a result, the surface temperature increased from 25 °C to over 90 °C, while the interior temperature exceeded 100 °C, resulting in the deactivation of enzymes present in rice bran (Figure 3). The activity of lipase in rice bran was quickly inhibited during the RF heating process [6]. Moreover, the moisture content of rice bran decreased from 14% to 4%.

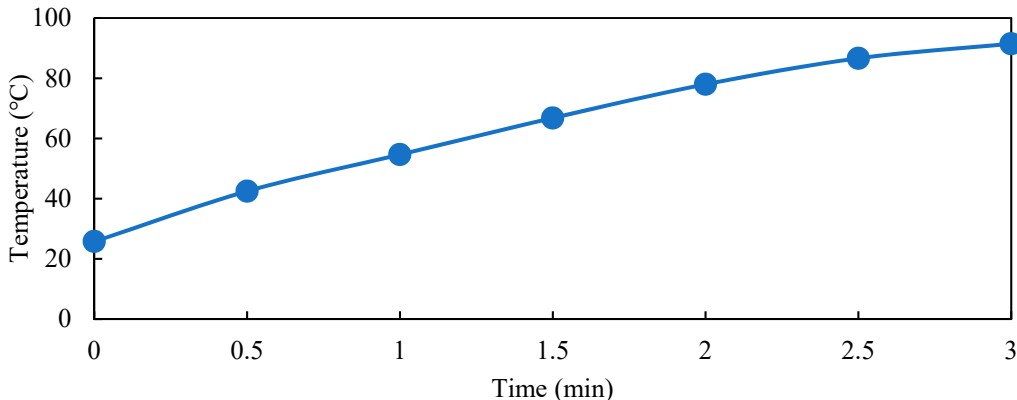

**Figure 3.** Surface temperature profile of 1 kg of rice bran during 5 kW HARF with an electrode gap of 10 cm. Data were expressed as mean ± S.D. (*n* = 3).

The moisture content of the ginseng residue from ultrasonic extraction by water was approximately 85%, resulting in a large loss factor that led the RF induction current to exceed the safety threshold of the equipment, posing a risk of failure. To solve this issue, 1 kg of RF-stabilized rice bran (with an initial moisture content of 4%) was mixed with varying quantities of ginseng residue (0.5, 1, 1.5, and 2 kg). This step aimed to reduce the moisture content of the samples, making them better suited for 5 kW, 40.68 MHz HARF drying. Figure 4 demonstrates how the application of 5 kW RF heating resulted in a significant increase in reflected RF power with sample loading. Furthermore, reducing the RF electrode gap led to an elevation in RF power. In order to effectively and safely execute RF drying for the four different combinations of rice bran and ginseng residue, the RF electrode gap of 14 cm was selected for HARF drying performance.

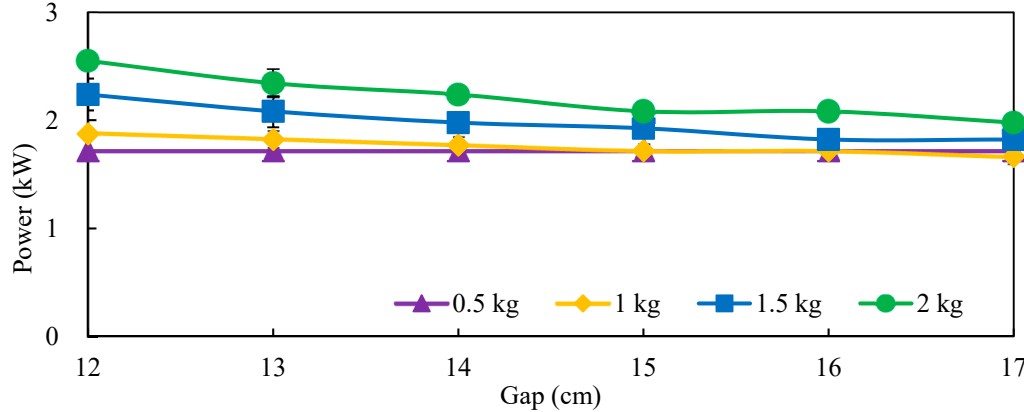

**Figure 4.** The power of 1 kg of rice bran mixed with different loadings of wet ginseng residue at 5 kW HARF with different electrode gaps. Data are expressed as mean ± S.D. (*n* = 3).

The 1 kg rice bran mixed with varying loads of wet ginseng residue (0.5, 1, 1.5, and 2 kg) exhibited different moisture contents of 31.67%, 45%, 53, and 58.33%, respectively. As a result, they possessed different loading weights of 1.5, 2, 2.5, and 3 kg for HARF drying. Figure 5 shows that temperatures quickly rose from 30 °C to 80 °C during the initial drying phases. At this time, sensible heat was provided, and then the temperature slowed down and became flat around 75~80 °C due to the conversion of energy into latent heat required for water evaporation and with 100 °C hot air to remove water vapor. Finally, the temperature of the samples reached 100 °C. Therefore, HARF could prevent microbial growth in wet ginseng residues from producing sour and spoiled odors due to a low temperature and long drying duration of only 45 °C cold air drying for 900 min.

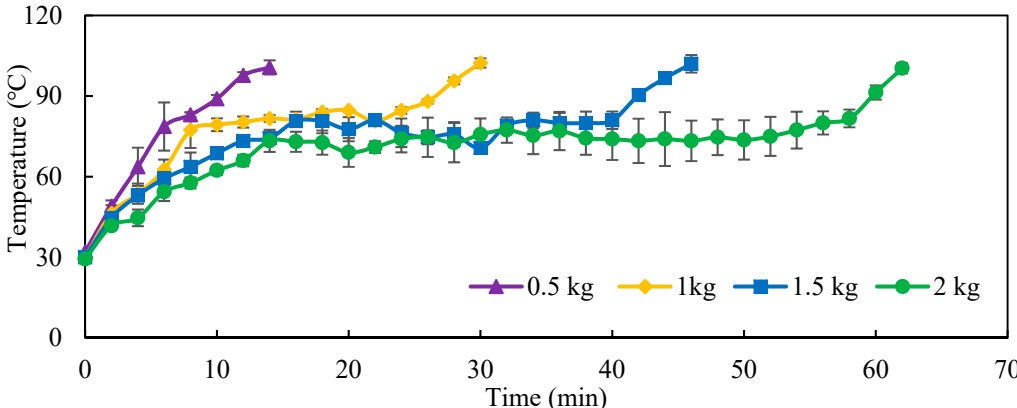

**Figure 5.** Temperature profiles of 1 kg of rice bran mixed with different loadings of wet ginseng residue during 5 kW HARF drying at an electrode gap of 14 cm. Data are expressed as mean ± S.D. (*n* = 3).

Huang et al. [18] discovered that both the mold and yeast counts in wild bitter gourd extract were dramatically reduced after 12 min of HARF drying. Moreover, RF heating required only 30 s to stop fermentation and 200 s to dry the 500 g solid-state fermented *Wolfiporia cocos* product, compared to 121 °C autoclaving for 60 min and 45 °C hot air drying for 180 min. There was no significant difference in the active components or antioxidant activity between HARF and traditional processes [29].

Figure 6 shows that the rate of decrease in moisture content (dry basis, g water/g dry material) was rather slow during the first 5 min due to the conversion of RF energy into sensible heat within the sample. Following this phase, a linear and constant rate drying period was observed. Notably, the rates of moisture content reduction were similar for all four samples. In the linear regression analysis of the drying curve, the drying rates for the mixtures of 1 kg of rice bran with 0.5, 1, 1.5, and 2 kg of ginseng residue ranged from approximately 26.2 to 27.4 g water/min (as shown in Table 1). HARF drying was effective, and the drying time was proportional to the sample loading.

**Table 1.** The drying rate of the 1 kg of rice bran (R) mixed with different loadings of wet ginseng residue (WG) during 5 kW HARF drying at gap of 14 cm.

| Total Weight (kg) | R + WG (kg) | Linear Regression Equation | Drying Rate (g/min) | $R^2$ |
|---|---|---|---|---|
| 1.5 | 1.0 + 0.5 | y = −27.423x + 1541.6 | 27.423 | 0.937 |
| 2.0 | 1.0 + 1.0 | y = −26.974x + 2101.7 | 26.974 | 0.967 |
| 2.5 | 1.0 + 1.5 | y = −26.325x + 2600.7 | 26.325 | 0.983 |
| 3.0 | 1.0 + 2.0 | y = −26.234x + 3132.1 | 26.234 | 0.990 |

y is the weight during RF drying, and x is the drying time.

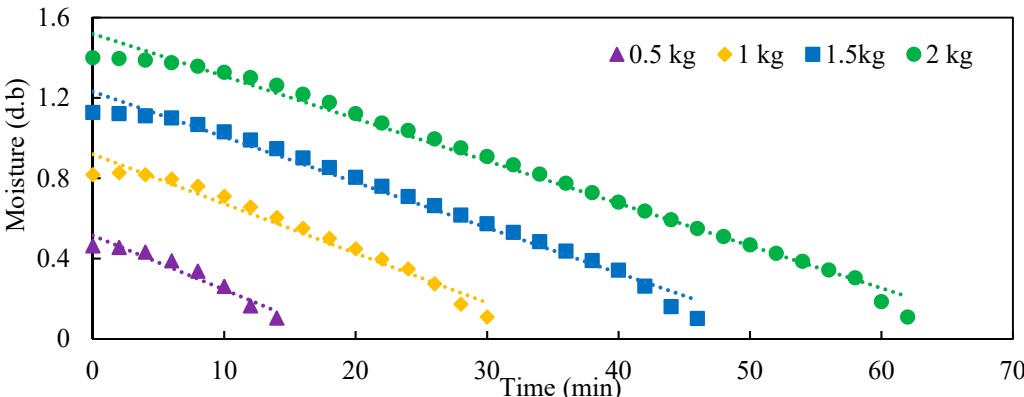

**Figure 6.** The drying curves of 1 kg of rice bran mixed with different loadings of wet ginseng residue during 5 kW HARF drying at an electrode gap of 14 cm.

Table 2 shows the estimated moisture content and drying time for 1 kg of rice bran mixed with 0.5, 1, 1.5, and 2 kg of ginseng residue. The moisture contents of these four mixtures increased when mixing more wet ginseng residue; however, they were less than 60%, far from the 85% in the original wet ginseng residue, making HARF drying difficult. The drying times to reduce the wet basis moisture content to 10% were estimated at 14, 30, 46, and 62 min, respectively. As a result, the energy consumption for drying 1 kg of wet ginseng residue ranged between 4.26 and 4.91 kWh/kg WG, with GHG carbon emissions ranging from 2.11 to 2.43 kg $CO_2$e/kg wet ginseng residue. In comparison, conventional air drying 1 kg of wet ginseng residue at 45 °C would require 900 min and 54.47 kWh/kg WG and emit 26.96 kg of $CO_2$e/kg to reduce the moisture content from 85% to 10%. Hence, employing HARF drying for wet ginseng residue could save up to 96% of the total time, achieve over 91% energy savings, and reduce carbon emissions.

The suitable 10 kW, 27.12 MHz HARF drying parameters were found to be an electrode gap of 8 cm, a thickness of 2 cm, 500 g of black carrot pomace, and a compaction density of 0.44 g/cm$^3$. The total drying times required to reduce the moisture content of black carrot pomace from 84% to 4% was 110 min, with an energy consumption of 3.06 kWh. In comparison to conventional hot air drying, HARF drying reduced the drying time by 39% [33]. Furthermore, an electrode gap of 9 cm and a 600 g carrot with a thickness of 2.5 cm were used as HARF (10 kW, 27.12 MHz) parameters to reduce carrot moisture contents from 88% to 8%, with a drying time of 60 min. When compared to hot air drying, HARF increased the drying rate while decreasing the drying time by 45% [34].

Before the extraction of peanut oil, peanuts were roasted at high temperatures to deactivate enzymes, reduce moisture content, create a distinctive flavor, and increase oil extraction efficiency [35]. Figure 7 shows the center temperature curves of 1 kg of peanuts during 4.5 min HARF (100 °C hot air-assisted 5 kW radio frequency with a 12 cm electrode gap). The final center temperature of peanuts increased in proportion to the heating duration, 120 °C, respectively. Evidently, the heating rate of the center temperature of peanuts was 21.341 °C /min. The roasted peanuts were then wrapped in parchment paper before being loaded into the oil press. Oil pressing was performed at 45 °C, and the oil yield was 39.52 ± 0.46%, while peanut residue accounted for about 60% of the initial peanut weight.

Almonds, with an initial moisture content of 8.47% (w.b.), were heated by HARF with a 10 cm electrode gap. The desired roasting temperature of 120 °C was achieved with 1 kg of almonds for 3.5 min, resulting in a moisture content of less than 2%. For comparison, 1 kg of almonds was roasted in a 105 °C conventional oven for 120 min [36]. In addition to the density and specific heat of the food, the heating rate of dielectric heating was mainly positively related to the dielectric loss factor of the food. However, the dielectric loss factor also changes as the temperature increases, and the moisture content decreases during the dielectric heating process [37].

**Table 2.** The drying time and energy consumption of the different proportions of rice bran and wet ginseng residue mixtures during 5 kW HARF drying at a gap of 14 cm and cold air drying.

| Drying Method | R+WG (kg) | Initial MC (%) | Drying Time to 10% MC (min) | Drying Time (min/kg WG) | Energy Consumption (kWh/kg WG) | Total GHG Emissions (kg $CO_2$e/kg WG) |
|---|---|---|---|---|---|---|
| Cold air | 0 + 1.0 | 85 | 900 | 900 | 54.47 ± 0.88 [a] | 26.96 ± 0.44 [a] |
| HARF | 1 + 0.5 | 31.67 | 14 | 28 | 4.26 ± 0.10 [b] | 2.11 ± 0.05 [b] |
| | 1 + 1.0 | 45 | 30 | 30 | 4.76 ± 0.25 [b] | 2.36 ± 0.12 [b] |
| | 1 + 1.5 | 53 | 46 | 31 | 4.91 ± 0.29 [b] | 2.43 ± 0.14 [b] |
| | 1 + 2.0 | 58.33 | 62 | 31 | 4.65 ± 0.22 [b] | 2.30 ± 0.11 [b] |

R: rice bran, WG: wet ginseng, MC: moisture content (wet basis, %), GHG: greenhouse gas. Data are expressed as mean ± S.D. (*n* = 3). [a,b] Means with different superscript letters in the same column indicate significant differences (*p* < 0.05).

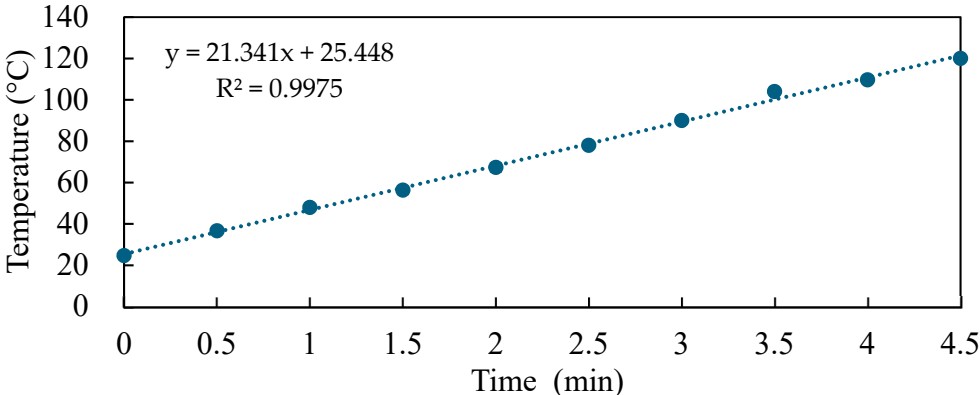

**Figure 7.** The center temperature curve of 1 kg of peanuts during 5 kW HARF roasting with a 12 cm electrode gap. Data are expressed as mean ± S.D. (*n* = 3), where y is the center temperature during RF roasting, and x is roasting time.

### 3.2. HARF Drying Hericium erinaceus Solid-State Fermented Rice Bran, Ginseng Residue, and Peanut Residue Product

According to a prior study, agricultural byproducts of rice bran and wet ginseng residue in a ratio of 2:3 were dried by HARF to achieve a ratio of rice bran to ginseng residue of 4:1, with the addition of peanut residue after oil expression. As a result, the 500 g solid-state media consisted of rice bran, ginseng residue, and peanut residue in a 4:1:1 ratio, with 45% moisture content in a bag. Sterilization and cooling were carried out, followed by a 5-week solid-state fermentation of *Hericium erinaceus* at 25 °C. Figure 8 shows that the mycelium of *Hericium erinaceus* grew in the bag from 1 to 5 weeks. In the second week, the white mycelium was noticeable, progressing to the appearance of mycelium during weeks 3 to 4. Over time, the mycelium gradually filled the bag.

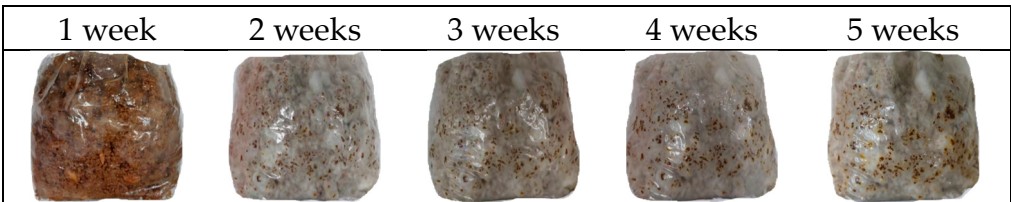

**Figure 8.** The picture of *Hericium erinaceus* solid-state fermentation using rice bran, ginseng residue, and peanut residue (4:1:1) as a medium for 1 to 5 weeks.

Then, 1 kg of *Hericium erinaceus* solid-state fermented product was sterilized and dried to reduce the moisture content to below 10% for storage by HARF (a 5 kW, 40.68 MHz

radio frequency heating system, assisted by 100 °C hot air). Figure 9 shows the drying and temperature curves during HARF, and this process achieved rapid drying within 28 min, reaching a temperature exceeding 100 °C, effectively pasteurizing the mycelium of *Hericium erinaceus*. Using the linear regression equation of the drying curve, the drying rate of HARF was determined to be 16.817 g water/min, with a high correlation coefficient ($R^2 = 0.985$).

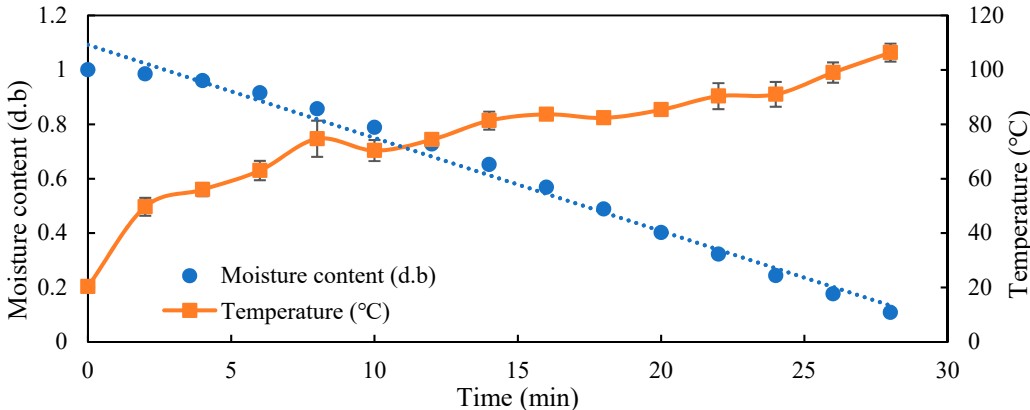

**Figure 9.** The drying and temperature curves of the *Hericium erinaceus* solid-state fermented product during 5 kW RF at a gap of 13 cm. Data are expressed as mean ± S.D. (*n* = 3).

Cultivation of Lion's Mane mushrooms typically involves either submerged or solid-state cultivation methods [38,39]. However, the sterilization, concentration, and drying processes downstream of submerged fermentation are time-consuming and costly. On the other hand, solid-state cultivation media are readily available and inexpensive, with lower costs for sterilization and drying as advantages [13,39]. Cheng et al. [38] performed *Hericium erinaceus* solid-state fermentation using various grains (polished rice, corn kernels, adlay, brown rice, and red beans) for 20-day cultivation; they were supported by mycelial growth and erinacine A production. Fan et al. [40] found that, while sawdust is a commonly used media for mushroom cultivation, due to forest logging limits, an agricultural byproduct is a good alternative medium. They utilized agricultural byproducts as *Hericium erinaceus* solid-state fermentation media, which resulted in higher total flavonoid content and better antioxidant activity in *Hericium erinaceus* fruiting bodies.

### 3.3. Whitening Effect of the Extracts from Hericium erinaceus Solid-State Fermented Product

Kojic acid is a widely used and regulatory-approved whitening agent; therefore, it was used as the positive control in this study. In Figure 10, the positive control group contained 100 μg/mL of kojic acid, as well as the water extracts from media R4G1, R4G1P1, and *Hericium erinaceus* solid-state fermented products obtained via RF-drying (HER4G1P1), and these extracts exhibited inhibition of tyrosinase activity at rates of 50.9%, 29.7%, 52.4%, and 50.7%, respectively. In comparison to the control group with 56% inhibition of tyrosinase activity, this indicated that the media R4G1P1 and fermented HER4G1P1 had potent tyrosinase inhibitory effects. Moreover, the raw agricultural byproducts obtained through solid-state fermentation by *Hericium erinaceus* showed greater inhibition of tyrosinase activity and an enhanced whitening effect because tyrosinase is a key enzyme in the synthesis of melanin. Tyrosinase activity can be inhibited with natural inhibitors, reducing melanin formation and resulting in a whitening effect. Similar results were found in the extract of *Ganoderma formosanum* mycelium, which also had a stronger inhibitory effect on tyrosinase [41].

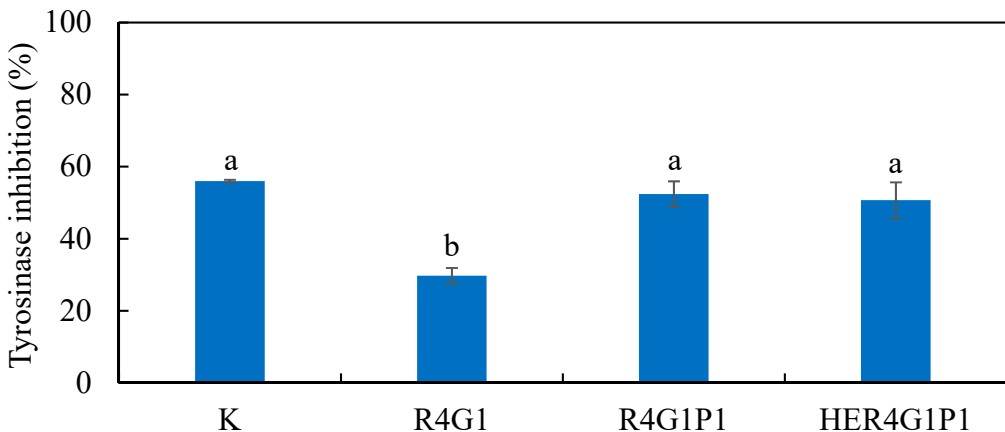

**Figure 10.** Effect of 100 μg/mL of water extracts in different concentrations on inhibition of tyrosinase activity. K: kojic acid; R4G1: water extracts from rice bran mixed with ginseng residues in a ratio of 4:1; R4G1P1: water extracts from the mixture of rice bran, ginseng residues, and peanut residue in a ratio of 4:1:1; HER4G1P1: water extracts from *Hericium erinaceus* solid-state fermented rice bran, ginseng residues, and peanut residue in a ratio of 4:1:1 product. Data are expressed as mean ± S.D. (*n* = 3). [a,b] Means with different letters in the samples indicate significant differences (*p* < 0.05).

Chen et al. [42] found that 107.6 mg/mL achieved 50% inhibition of tyrosinase activity. Chen et al. [43] discovered that rice bran *Lactobacillus* fermentation broth could produce L-DOPA with an IC50 of 9.23 mg/mL. This inhibitory effect was found to be even stronger than that of arbutin (with an IC50 concentration of 0.52 mg/mL). The protocatechuic acid methyl ester from the extracts of black rice bran inhibited 75.4% of tyrosinase activity at 0.5 μmol/mL [44]. The IC50 values of the extract from defatted rice bran were 0.99 mg/mL for monophenolase activity and 1.92 mg/mL for diphenolase activity, which were equivalent to the standard arbutin with no significant difference [45].

The skin-whitening effects of zebrafish embryos were studied. Initially, melanin area analysis using a computer image was performed, as shown in Table 3. In the control group, zebrafish embryos immersed in water showed a 0.423 mm$^2$ melanin area. When the melanin areas of zebrafish embryos images taken at 78 hpf were compared to the control group, it was observed that 100 μg/mL kojic acid, as well as the extracts from the media R4G1, R4G1P1, and fermented HER4G1P1, reduced in the melanin area on the zebrafish embryo surface by 24.74%, 21.57%, 40.20%, and 58.03%, respectively. These findings indicated that these extracts from agricultural byproducts had the ability to prevent melanin production. Further, *Hericium erinaceus* solid-state fermented products using a 4:1:1 ratio of rice barn, ginseng residue, and peanut residue as a medium increased whitening effect.

Jin et al. [2] found that 40 μM of ginsenosides Rg5 and Rk1 significantly inhibited α-MSH-induced tyrosinase expression in B16F10 murine melanoma cells. Zebrafish embryos were exposed to concentrations of 200 μg/mL of black ginseng extract from 10 to 72 hpf, resulting in a significant decrease in tyrosinase expression. Indeed, the ginsenosides Rk1 and Rg5 drastically reduced tyrosinase activity and expression levels. The stimulation of the MEK-ERK signaling pathway was found to be the primary mechanism of their anti-melanogenic effects. Ginsenoside Rk1 and Rg5 from black ginseng decreased tyrosinase expression and, via activation of MEK-ERK signaling, negatively regulated melanogenesis by degrading MITF.

**Table 3.** Whitening effect of 100 μg/mL of water extracts from media and *Hericium erinaceus* solid-state fermented product.

| Sample | Area (mm$^2$) | Relative Reduction (%) | Image |
|---|---|---|---|
| Control | 0.423 ± 0.015 [a] | |  |
| K | 0.319 ± 0.014 [b] | 24.74 ± 3.24 [c] |  |
| R4G1 | 0.332 ± 0.017 [b] | 21.57 ± 4.13 [c] |  |
| R4G1P1 | 0.253 ± 0.006 [c] | 40.20 ± 1.31 [b] |  |
| HER4G1P1 | 0.178 ± 0.005 [d] | 58.03 ± 1.10 [a] |  |

K: kojic acid; R4G1: water extracts from rice bran mixed with ginseng residues in a ratio of 4:1; R4G1P1: water extracts from the mixture of rice bran, ginseng residues, and peanut residue in a ratio of 4:1:1; HER4G1P1: water extracts from *Hericium erinaceus* solid-state fermented rice bran, ginseng residues, and peanut residue product. Data are expressed as mean ± S.D. ($n = 3$). [a–d] Means with different superscript letters in the same column are significantly different ($p < 0.05$).

## 4. Conclusions

This study used a 5 kW HARF system to blanch rice bran, dry ginseng residue, roast peanut, and pasteurize and dry solid-state fermented products. Heating 1 kg of rice bran heated for 3 min elevated the temperature to above 90 °C, thus deactivating the enzymes in rice bran. The drying rate of 1 kg rice bran combined with ginseng residue was approximately 27 g/min, resulting in a rapid moisture reduction to less than 10%. Compared to cold air drying, this method saved around 96% of the total time and 91% of the energy. Moreover, before expressing peanut oil, 1 kg of peanuts was roasted by HARF for only 4.5 min to reach 120 °C. Finally, 1 kg of *Hericium erinaceus* 5-week solid-state fermented product (using rice bran, ginseng residue, and peanut residue in a 4:1:1 ratio as a medium) was pasteurized and dried by HARF for 28 min. The 100 μg/mL of water extract from media and *Hericium erinaceus* solid-state fermented product reduced the relative melanin area in the zebrafish embryo. Both RF and microwave heating utilize dielectric heating; however, RF penetrates deeper than microwaves, resulting in more uniform heating when compared to microwaves and hot air drying. RF heating food can reduce the processing time and energy consumption, thus lowering the cost of reusing agricultural by-products. Moreover, further fermentation technology can be employed to enhance bioactive substances. This will surely lead to more agricultural by-products being reused with large-scale RF equipment.

**Author Contributions:** Conceptualization, supervision, writing, and project administration, S.-D.C.; analysis and assisting with fermentation and radio frequency experiments, C.-L.Y.; and whitening experiments and original draft preparation, Z.-Y.L. All authors have read and agreed to the published version of the manuscript.

**Funding:** This research received no external funding.

**Data Availability Statement:** The data and samples presented in this study are available on request from the corresponding author. Data are contained within the article.

**Acknowledgments:** We would like to thank Bor-Yann Chen for giving our suggestions for the manuscript; ACEXTRSCT Biotechnology Co., Ltd. for providing wet ginseng residues; and the Minfeng Organic Farm for providing rice bran for this study.

**Conflicts of Interest:** The authors declare no conflicts of interest.

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
