# Peer review of "Study on Radio Frequency-Treated Agricultural Byproducts as Media for Hericium erinaceus Solid-State Fermentation for Whitening Effects"

_processes, doi:10.3390/pr12040830_

Round 1
Reviewer 1 Report
Comments and Suggestions for Authors
The submitted manuscript for review titled "Study on Radio Frequency Treated Agricultural Byproducts as Media for Hericium erinaceus Solid-State Fermentation for Whitening Effects" presents an intriguing exploration of the use of hot air – assisted radio frequency to dry rice bran, ginseng residue, roast peanut. The experimental results of this work is significant strength; however, upon reviewing the manuscript, the reviewer requests clarification on several issues:
- In section 2.2, the schematic figures or pictures of the equipment system should be shown.
- The regression equations of drying kinetic are written under function of y = f(x), the expression the meaning of y and x in those equations should be explained. Why do trends of drying kinetics and temperature change linear and constant for almost drying process?
- Tables of nomenclature and abbreviations should be added.
- How to evaluate the experimental errors?
Author Response
Thank your sugession, I answer the following question. The revised manuscript is upload.
(1) In section 2.2, the schematic figures or pictures of the equipment system should be shown.
Answer: We add “A hot air-assisted radio frequency (HARF) equipment (as shown in Figure 1) (5 kW, 40.68 MHz, 220V, Yh-Da Biotech Co., Ltd., Yilan, Taiwan).”
(2) The regression equations of drying kinetic are written under function of y = f(x), the expression the meaning of y and x in those equations should be explained. Why do trends of drying kinetics and temperature change linear and constant for almost drying process?
Answer: We add “Where y is the weight during RF drying, and x is drying time.” in the end of Table 1.
In the early stage of radio frequency drying, the temperature of the mixture of rice bran and ginseng residue rises rapidly. At this time, sensible heat is provided, and then the temperature slows down and becomes flat. This has converted the energy into latent heat required for water evaporation, and with the help of 100°C hot air The evaporated water vapor can be blown away, causing the moisture content to decrease linearly.
(3) Tables of nomenclature and abbreviations should be added.
Answer: Because including a list table of nomenclature and abbreviations would change the overall layout, they are still highlighted in the manuscript when they first appear.
(4) How to evaluate the experimental errors?
Answer: Each set of analyzes was performed at least three times.
Reviewer 2 Report
Comments and Suggestions for Authors
File attached

Need minor editing at few places
Author Response
Thank your suggestion. We answer the following questions. The revised manuscript is upload.
1. Introduction part need to be reduced.
Answer: The introduction part was reduced from original line 29-149 to line 29-123.
2. Line no. 212 Replace ‘could be’ by ‘was’
Answer: We have replaced ‘could be’ by ‘was’ in Line no. 217.
3. Line 217-220 Rewrite the sentence giving formula in equation form.
Answer: We add to read current (A) and calculate the RF power using the following equation: …………………(1)
Equations (2-5) has been written in the Equation editor.
4. Line 264: Relace ‘extract’ be ‘extraction’
Answer: We have changed ‘extract’ be ‘extraction’ .
5. Experimental design is missing
Answer: We add “2.3. Experimental design” flow chart.
2.3. Experimental design
The experimental design diagram above showed that it was separated into two parts. First, the drying rate and energy consumption of HARF drying ginseng residue and rice bran combinations were assessed. The HARF-treated rice bran and ginseng residue mixture, and peanut residue after oil extraction were then mixed in a 4:1:1 ratio (45% moisture content) to make a solid-state fermented medium. The water extracts from 5-week Hericium erinaceus solid-state fermented products were analyzed the whitening effect.
6. Line 412-435: Paragraph is focusing more on review rather than results and should be shifted. It is advised to add this as justification with results but in brief with proper reference.
Answer: We modified them in paragraph.
7. Lint 459 onwards: Start the paragraph with the results and add this general matter as justification.
Answer: We modified them in paragraph.
8. Table data missing SD values – must be added(confirmation)
Answer: We added SD in Energy consumption (kWh/ kg WG) and Total GHG emissions (kg CO2e/ kg WG) in Table 2. We added SD in relative reduction (%) in Table 3.
Reviewer 3 Report
Comments and Suggestions for Authors
The study presents a current topic. The main aim is formulated appropriately. A study aimed at increasing the utilization of agricultural byproducts is very important from a practical point of view. Presented information are beneficial for this research field. The title of the article is a bit long, but it sufficiently reflects content. The abstract and key words are informative. The figures and tables adequately complement the presentation of the results. The methodology is described in an appropriate manner. The manuscript has a very good scientific level and most of the comments and specific recommendations are formal.
1. Equations (1 – 4) should be written in the Equation editor.
2. The writing and use of physical quantities units should be checked throughout the text of the manuscript.
3. Formally, it is necessary to unify the writing of numerical values and units of physical quantities with a gap.
4. In the equations listed in Table 1 and Figure 5, I recommend using a specific designation of physical quantities instead of x, y.
5. Abbreviation of physical units should be corrected.
6. I recommend to supplement "Conclusions" with novelties, more universally formulated benefits and presentation of the applying possibilities of the HARF method to other samples of biological materials.
7. "References": Cited publications contain in many cases the names of some of the authors of the assessed manuscript. To improve the possibility of international response, I recommend adding alternative citations of publications by other foreign authors.
GENERAL JUDGEMENT: The paper is acceptable for publication after minor revision.

Author Response
Thank your sugession. We answer the following question. The revised manuscript is upload.
1. Equations (1-4) should be written in the Equation editor.
Answer: We added to read current (A) and calculate the RF power using the following equation: (1), and equations (2-5) has been written in the Equation editor.
2. The writing and use of physical quantities units should be checked throughout the text of the manuscript.
Answer: We checked the physical quantities units throughout the text of the manuscript.
3. Formally, it is necessary to unify the writing of numerical values and units of physical quantities with a gap.
Answer: We checked and corrected them.
4. In the equations listed in Table 1 and Figure 5, I recommend using a specific designation of physical quantities instead of x, y.
Answer: We add “Where y is the weight during RF drying, and x is drying time.” in the end of Table 1. Moreover, we added figure 1, and Figure 5 was changed to Figure 6, “Where y is the center temperature during RF roasting, and x is roasting time.” in the end of Figure 6.
5. Abbreviation of physical units should be corrected.
Answer: We checked the physical quantities units throughout the text.
6. I recommend to supplement "Conclusions" with novelties, more universally formulated benefits and presentation of the applying possibilities of the HARF method to other samples of biological materials.
Answer: We removed “Radio frequency can be successfully used in agricultural byproducts to save both time and energy.” in the end of conclusion, and added the following sentences: “Both radio frequency and microwave heating utilize dielectric heating; however, radio frequency penetrates deeper than microwaves, resulting in more uniform heating when compared to microwaves and hot air drying. RF heating food can reduce processing time and energy consumption, thus lowering the cost of reusing agricultural by-products. Moreover, further fermentation technology is employed to enhance the bioactive substances. It will surely lead to more agricultural by-products being reused in coupled with large-scale RF equipment.”
7. References: Cited publications contain in many cases the names of some of the authors of the assessed manuscript. To improve the possibility of international response, I recommend adding alternative citations of publications by other foreign authors.
Answer: The primary agricultural by-products utilized in this study consist of rice bran post-milling, ginseng residue following ultrasonic extraction, and peanut residue post-pressing for peanut oil, they are predominantly associated with oriental food industry. Moreover, the prevalence of Hericium erinaceus fermentation is notably higher in Eastern regions, reflecting the majority of citations originating from Asian researchers, accounting for nearly two-thirds of the total references. Notably, China has emerged as a prolific source of publications concerning radio frequency heating in recent years. In the future, we plan to investigate widely-utilized agricultural by-products globally, fostering sustainable development while enhancing their application value.